# Low Flow versus No Flow: Ischaemia Reperfusion Injury Following Different Experimental Models in the Equine Small Intestine

**DOI:** 10.3390/ani12162158

**Published:** 2022-08-22

**Authors:** Anna Marei Grages, Nicole Verhaar, Christiane Pfarrer, Gerhard Breves, Marion Burmester, Stephan Neudeck, Sabine Kästner

**Affiliations:** 1Clinic for Horses, University of Veterinary Medicine Hannover, 30559 Hannover, Germany; 2Institute for Anatomy, University of Veterinary Medicine Hannover, 30559 Hannover, Germany; 3Institute for Physiology and Cell Biology, University of Veterinary Medicine Hannover, 30559 Hannover, Germany; 4Small Animal Clinic, University of Veterinary Medicine Hannover, 30559 Hannover, Germany

**Keywords:** intestine, ischaemia, reperfusion, apoptosis, inflammation, colic, horse

## Abstract

**Simple Summary:**

One of the main causes of colic in horses is the occlusion of the intestinal blood vessels after displacement or entrapment of the small intestine. In search of new therapies to treat this lethal disease, experimental models have been used to simulate the clinical situation. Both low flow (LF) models with partial blood flow occlusion as well as no flow (NF) models with complete occlusion have been implemented in different studies. This has led to conflicting results and comparative studies are lacking. The objective of this study was to characterize the development of intestinal injury over time in two different experimental models implementing either partial or complete vessel occlusion. Under general anaesthesia, local intestinal blood flow was reduced by 80% in seven horses (LF), and by 100% in another seven horses (NF). The LF group exhibited more bleeding in the intestinal wall and a relatively high variability in intestinal oxygen levels and tissue damage. The NF group showed lower oxygen levels and decreased barrier function of the intestinal wall. These results aid in the selection of the suitable experimental model for future studies. The high variability following LF suggests that an NF model may produce more consistent intestinal damage.

**Abstract:**

In experimental studies investigating strangulating intestinal lesions in horses, different ischaemia models have been used with diverging results. Therefore, the aim was to comparatively describe ischaemia reperfusion injury (IRI) in a low flow (LF) and no flow (NF) model. Under general anaesthesia, 120 min of jejunal ischaemia followed by 120 min of reperfusion was induced in 14 warmbloods. During ischaemia, blood flow was reduced by 80% (LF, *n* = 7) or by 100% (NF, *n* = 7). Intestinal blood flow and oxygen saturation were measured by Laser Doppler fluxmetry and spectrophotometry. Clinical, histological, immunohistochemical and Ussing chamber analyses were performed on intestinal samples collected hourly. Tissue oxygen saturation was significantly lower in NF ischaemia. The LF group exhibited high variability in oxygen saturation and mucosal damage. Histologically, more haemorrhage was found in the LF group at all time points. Cleaved-caspase-3 and calprotectin-stained cells increased during reperfusion in both groups. After NF ischaemia, the tissue conductance was significantly higher during reperfusion. These results aid in the selection of suitable experimental models for future studies. Although the LF model has been suggested to be more representative for clinical strangulating small intestinal disease, the NF model produced more consistent IRI.

## 1. Introduction

In horses, intestinal ischaemia reperfusion injury (IRI) due to strangulating intestinal disease is associated with high mortality rates [1,2]. To investigate the pathophysiology and potential therapeutic interventions for intestinal strangulations in horses, experimental models simulating intestinal IRI have been used for experimental in vivo trials. These models can differ in the type and duration of ischaemia applied, as well as the methodology of vascular occlusion.

In small intestinal ischaemia models, segmental jejunal ischaemia has been created by occluding the mesenteric vessels with ligatures [3,4,5,6,7] or haemostatic clamps [8,9,10,11]. For the duration of ischaemia, 90 to 120 min has frequently been elected [5,9,10,11,12,13,14,15]. However, reperfusion times vary markedly between the different studies, and sequential measurements during ischaemia and reperfusion are lacking. Differing degrees of vascular occlusion have been used, as it remains unclear what type of ischaemia most accurately represents the clinical situation. Consequently, ischaemia models with a complete elimination of intestinal blood flow (no flow, NF), and models with varying degrees of blood flow reduction (low flow, LF) have been implemented [3,4,8,10,11,12,14,16,17,18,19,20].

Monitoring blood flow before and after induction of ischaemia and at the onset of reperfusion is of great importance to ensure that the previously defined ischaemia is achieved in all subjects, thereby warranting comparable and reproducible results [21]. Previous studies have used surface oximetry [8], Doppler sonography [9,17], blood pressure measurement [9], Laser Doppler fluxmetry (LDF) [19] or LDF in combination with microlightguide spectrophotometry (MS) [5,6,15]) for this purpose. 

Due to the variation in the type of experimental ischaemia applied, the comparability of the results between different studies is limited, and inconsistent results have been reported. One point of discussion is the occurrence of reperfusion injury, with conflicting results regarding mucosal deterioration [3,5,8,9,15] or the increase in apoptotic cells [7,11,15] following LF or NF ischaemia during reperfusion. These discrepancies complicate the interpretation and hinder the extrapolation to the clinical patient.

Supported by these limitations of in vivo models, we should aim to reduce the number of animal experiments and even strive for its complete replacement. In vitro techniques such as cell cultures [22] and organoids [23,24] are gaining popularity and can replace certain in vivo studies. Nevertheless, at this point the complexity of strangulating intestinal diseases cannot be fully reproduced by the use of organoids, and comparative studies are lacking. In vivo studies may still be indicated in certain circumstances, for example for the clinical application of new therapeutic strategies that aim to improve the survival of horses with intestinal ischaemia. However, care should be taken that controlled and reliably reproducible ischaemia models are used, in order to optimize the results of in vivo studies and to keep the number of subjects to a minimum.

The objective of this study was to characterize and compare the development of intestinal ischaemia reperfusion injury over time in two different experimental models implementing either LF or NF ischaemia. We hypothesized that NF ischaemia would elicit more severe mucosal damage during ischaemia. The second hypothesis was that LF ischaemia would exhibit increased inflammation and that there would be a significant progression of intestinal injury during reperfusion. Furthermore, we hypothesized that LF ischaemia would be associated with a higher variability of the intestinal blood supply within the ischaemic segment.

## 2. Materials and Methods

### 2.1. Animals

The study was reviewed by the Ethics Committee for Animal Experiments of Lower Saxony, Germany, and approved according to §8 of the German Animal Welfare Act (LAVES 33.8-42502-04-19/3240). Fourteen horses, owned by the University of Veterinary Medicine Hannover, were divided into 2 groups using simple randomization with an equal allocation ratio. One group (*n* = 7) underwent NF ischaemia while the other group (*n* = 7) was subjected to LF with 80% occlusion of the mesenteric blood flow. The 14 warmblood horses had a mean age of 17 ± 7 years and weighed 554 ± 69 kg. There were 8 mares, 3 geldings and 3 stallions, without any differences between the groups. The horses were elected for euthanasia due to severe musculoskeletal problems. According to the physical examination, blood cell counts and faecal egg count, the horses were judged to be systemically healthy. They were stabled at the facilities of the equine clinic of the University of Veterinary Medicine Hannover at least 2 weeks prior to surgery, with free access to hay and water and were hand walked daily. On the day of the trial, food but not water was withheld for 6 h prior to anaesthesia.

### 2.2. Surgical Procedure

After premedication with dexmedetomidine 5 µg/kg (Dexdomitor, Orion Corporation, Espoo, Finland), induction of general anaesthesia was performed with 0.05 mg/kg diazepam (Diazedor, WDT eG, Garbsen, Germany) and 2.2 mg/kg ketamine (Narketan, Vétoquinol GmbH, Ismaning, Germany). Maintenance of anaesthesia was implemented with isoflurane (Isofluran CP, CP-Pharma GmbH, Burgdorf, Germany) in 100% oxygen combined with a continuous rate infusion of dexmedetomidine (5 µg/kg/h). Mean arterial blood pressure was monitored by direct pressure measurements in the facial artery and maintained between 60 and 80 mmHg by administrating lactated Ringer’s solution (Ringer-Laktat EcobagClick, B. Braun Melsungen AG, Melsungen, Germany) and dobutamine (Dobutamin-ratiopharm 250 mg, Ratiopharm GmbH, Ulm, Germany) to effect. Blood gas analysis was performed every 30 min, and the horses were mechanically ventilated using intermittent positive pressure ventilation.

The horses were placed in dorsal recumbency, and the ventral midline was aseptically prepared. Subsequently, a routine pre-umbilical median laparotomy was performed. The intestinal segment that served as control sample during pre-ischaemia (P) and after reperfusion (RC) was located 7 m oral to the ileocecal fold. Intestinal microperfusion (in arbitrary units, AU), tissue oxygen saturation (StO_2_) and haemoglobin (Hb) were measured in the different intestinal segments by use of Laser Doppler fluxmetry and microlightguide spectrophotometry (LDFMS) with a commercially available device (O2C, LEA Medizintechnik GmbH, Giessen, Germany). Following the control measurements, segmental ischaemia was induced in 2 m of jejunum that was located 1 m oral to the ileocaecal fold by occlusion of the mesenteric arteries and veins with umbilical tape. In the horses subjected to NF ischaemia, each jejunal vessel was also occluded with a haemostatic clamp to ensure 100% blood flow reduction. The LF ischaemia was induced by tightening the umbilical tape under monitoring LDFMS until arterial blood flow was reduced with 80%. Ischaemia was maintained for 120 min, followed by 120 min of reperfusion. The ischaemic segment was divided into 4 sub-segments, each supplied by its own arcade vessel. Hourly LDFMS measurements were performed in each sub-segment (Figure 1). A tissue sample was also taken each hour from a single sub-segment in an aboral to oral direction. Directly following the last measurement, the horses were euthanized with 90 mg/kg pentobarbital administered intravenously (Release 50 mg/ml, WDT eG, Garbsen, Germany), and transferred to the institute of anatomy for educational purposes.

### 2.3. Clinical Evaluation of the Intestine

During the experiment, the intestine was evaluated by manual palpation for wall thickness with a semiquantitative score by 2 observers (Table 1).

The intestinal colour as seen from the serosal side (from now on referred to as serosal colour) was judged from intraoperative photographs (Canon Legria HF G10, Canon Deutschland, Krefeld, Germany) that were taken during each experimental phase under the same lighting conditions. The images were graded using this semiquantitative score by 2 observers that were blinded for the identity of the photographs (Table 1, Figure 2).

### 2.4. Histology and Immunohistochemistry

The tissue samples were fixed in formalin and embedded in paraffin. Subsequent to routine processing, the sections were stained with haematoxylin and eosin (H&E). One section per sample was evaluated by light microscopy (AXIO Scope.A1, Carl Zeiss GmbH, Oberkochen, Germany) by a single observer who was unaware of the identity of the slides. Ten villi were evaluated for epithelial separation (EPS) and haemorrhage (HS) using a modified Chiu score (Table 2). The EPS scores of the 10 villi were averaged for the slide, and the HS was scored for the complete section.

Immunohistochemical staining for cleaved-caspase-3 (dilution 1:200, rabbit-anti-human, CleavedCaspase-3Asp175 antibody, Cell Signalling Technology Europe B.V., Leiden, The Netherlands) as a marker for apoptosis was performed as described previously [15]. Furthermore, inflammatory cells were identified by immunohistochemical staining for cytosolic calprotectin (monoclonal mouse anti-human myeloid/histiocyte antigen, clone MAC 387, DakoCytomation, Glostrup, Denmark), representing mainly neutrophils, monocytes and macrophages [26].

By use of a microscope camera and accompanying software (Axiocam 105 color and Software ZEN 2.3, Carl Zeiss GmbH, Oberkochen, Germany), the cleaved-caspase-3 positive cells were counted in the mucosa and expressed in cells/mm^2^ following exact surface measurements. The calprotectin positive cells were counted separately in the mucosa, submucosa, muscularis and serosa using the same technique.

### 2.5. Electrophysiology

Electrophysiological measurements were performed on mucosal samples of 12 horses. The samples of 1 horse from each group could not undergo this analysis, due to the limited availability of the Ussing chambers during the experiment.

The Ussing chamber experiment was performed as previously described [6,27]. In brief, the mucosa was manually stripped of the seromuscular layer and mounted in Ussing chambers, with 1.13 cm^2^ of tissue exposed. The tissue was placed in a modified Krebs–Henseleit buffer aerated with carbogen. Short circuit currents (Isc in μEq/cm/h) and transepithelial potential differences (PDt) were measured using a computer-controlled voltage clamp device (K. Mußler, Aachen, Germany). Tissue conductance (Gt) was determined from the changes in PDt following bipolar current pulses of 100 μA/cm^2^. Measurements were performed in the samples P, I2 and R2 under basal conditions. Furthermore, 10 mM alanine and 10 mM glucose were added to the luminal side of different chambers of all tissue samples, to investigate the transport of both nutrients. 10^−5^M forskolin (Sigma Aldrich, Darmstadt, Germany) was added to the serosal side at the end of the experimental period to confirm tissue viability. For statistical analysis of the tissue conductance, the results of the alanine and glucose chambers were pooled.

### 2.6. Data Analysis

A power analysis was performed prior to commencing the study (G*Power 3.1.9.1^s^). With a standard deviation of 0.3, based on a power of 0.8 and alpha of 0.05, using a Mann–Whitney Test, 7 horses per group were determined necessary to detect a difference of 0.5 grade in the histomorphology score between the different models. This analysis was also performed for the immunohistochemistry cell counts, yielding a comparable number of horses per group, depending on the estimated values of the standard deviation. 

Statistical analysis and graph design were performed using commercially available software (Graphpad Prism 8.3, Graphpad Software Inc., San Diego, CA, USA). The distribution of the data was visually assessed in the qq-plots of the model residuals. Furthermore, a Shapiro–Wilks test was performed. Levene’s test and visual assessment of the homoscedasticity plots was used to evaluate the variance homogeneity. Normally distributed variables were expressed as mean (± standard deviation), and the non-parametric as median (min-max). *p*-values of <0.05 were considered significant. Categorical variables were displayed as frequency distribution.

For analysis of the normally distributed data, a repeated measures two-way analysis of variance (ANOVA) was performed for 1 independent (group) and 1 repeated (time point) effect. The different time points and groups were compared, with the horses as subject effect. The *p*-values were subjected to the Greenhouse–Geisser correction. Post-hoc multiple pairwise comparisons between the groups were performed for each time point with Sidak’s test. Post-hoc multiple pairwise comparisons to compare the time points with the pre-ischaemia sample were conducted with a Dunnett’s test. For the normally distributed variables, the coefficient of variation (CV) was also documented.

Data that were not normally distributed were analysed with non-parametric tests. The results between the different models were compared at each time point using a Mann–Whitney-U Test. For comparing the correlated different time points, a Friedman test for repeated measures was used, with multiple pairwise comparisons by use of a post-hoc Sidak test. The frequency distribution of the categorical variables was compared between the 2 different models at each time point using a chi-squared test.

To investigate the variability of the blood supply between the different sub-segments supplied by different arcades within one ischaemic segment of each single horse, the StO_2_ of these sub-segments was evaluated. For each individual horse, the oxygen saturation measurements of the different sub-segments during 60 and 120 min ischaemia were used to determine the range and CV of StO_2_ within the ischaemic segment of this single horse. These results were then averaged per group (LF or NF) to determine the variability of the sub-segments for each type of ischaemia.

## 3. Results

### 3.1. Intestinal Microperfusion, Oxygen Saturation and Haemoglobin

Pre-ischaemia, the mean value of intestinal blood flow was 202 (±49) AU in the LF group and 226 (±74) AU in the NF group (Figure 3). This was reduced with the induction of ischaemia to 79 (±36) in the LF group and to 83 (±47) AU in the NF group. After 1 h of ischaemia, 53 (±23) AU were measured in the LF group and 51 (±44) AU in the NF group, and after 2 h of ischaemia this was 48 (±16) AU and 46 (±16) AU, respectively. Following reperfusion, blood flow increased above the baseline level (Figure 3).

In both groups, blood flow was significantly reduced during ischaemia, with no difference between LF and NF. In the NF group, all relevant vessels were additionally occluded with vascular clamps, yet nevertheless a blood flow measurement of 0 AU could not be measured at any time.

The application of ischaemia reduced the StO_2_ compared to baseline in both groups, yet this was significantly more in the NF group compared to the LF group (Figure 4). After 1 h and 2 h of ischaemia, a StO_2_ of 33 (±19)% and 35 (±17)% was measured in the LF group, while the NF group had a saturation of 8 (±2)% and 6 (±1)%. The CV at 60 and 120 minutes of ischaemia were 59% and 49% in the LF group, and 24% and 21% in the NF group. During reperfusion, StO_2_ increased above baseline levels in both groups. 

Table 3 illustrates the variability in StO_2_ of the different sub-segments within one ischaemic intestinal segment of a single horse, averaged per group. The range between the different sub-segments within one single segment had an average of 40.3% oxygen saturation in horses of the LF group, indicating a high variability of oxygenation within a single segment during this type of ischaemia, whereas the NF group showed a range of 10%. Contrarily, the coefficient of variation was comparably high in both groups. 

Baseline measurement of intestinal Hb showed 63 (±5) AU in the LF group and 68 (±15) AU in the NF group (Figure 5). Following induction of ischaemia and after 1 h and 2 h of ischaemia, an Hb of 71 (±12), 79 (±6), and 85 (±5) AU was measured in the LF group, respectively. The NF group exhibited 64 (±8), 65 (±11) and 65 (±7) AU Hb during these time points. After 2 h of ischaemia, a significantly higher Hb value was present in the LF group compared to the NF group (Figure 5). During reperfusion, the values of the LF group hardly changed, yet in the NF group the values increased significantly during reperfusion.

### 3.2. Clinical Observations

During ischaemia, a clear change in wall thickness and serosal colour was seen in both groups compared to pre-ischaemia (Table 4, Figure 6). In the group comparison, serosal colour and wall thickness of the LF and NF groups differed significantly from each other with a greater increase in wall thickness in the LF group and a different serosal colour grade in the NF group (Table 4). During reperfusion, both groups showed serosal colour grade C and a decreasing trend in wall thickness.

### 3.3. Mucosal Histomorphology

Pre-ischaemia, none of the intestinal samples had any epithelial separation (EPS of 0), which increased in both groups during ischaemia. In the LF group, this did not reach significant levels until reperfusion. Furthermore, a high variability could be noted between the different horses subjected to LF, as illustrated by the wide boxes of the box plots (Figure 7a). The horses of NF group showed less variability as reflected by the narrow box of the box plot, and a significant increase in epithelial separation could already be seen after 2 hours of ischaemia as well as during reperfusion. There was no significant progression of intestinal damage during reperfusion, and there were no significant differences between the groups.

During ischaemia, HS increased significantly in both groups compared to pre-ischaemia (Figure 7b), yet the LF group had a significantly higher HS than the NF group during ischaemia and reperfusion (I1 *p* < 0.01, I2, R2 *p* < 0.001, R1 *p* < 0.05).

### 3.4. Apoptosis

In both groups, there was only a very slight increase in apoptotic cells during ischaemia. After 1 h and 2 h of reperfusion, the number of cleaved caspase-3 positive cells increased significantly in both groups. This response appeared more pronounced after NF ischaemia, yet without statistically significant differences between the groups (Figure 8).

### 3.5. Inflammation

In all intestinal layers, there was a slight increase of calprotectin-positive cells during ischaemia compared to pre-ischaemia, which was only significant in the mucosa and submucosa in the LF group after 2 h of ischaemia (Figure 9). During reperfusion, significant inflammatory cell infiltration occurred in all intestinal layers compared to pre-ischaemia. The remote control sample at reperfusion showed significantly more calprotectin positive cells in the muscularis of the LF group and in the serosa of the NF group compared to the pre-ischaemic control segment.

Comparing the groups, there was more inflammatory cell infiltration in the serosa during ischaemia in the LF group than in the NF group. After 2 h of reperfusion, the NF group showed more calprotectin positive cells in the submucosa than the LF group (*p* = 0.04).

### 3.6. Electrophysiology

In the pre-ischaemia samples, there was a significant response in short circuit current to the addition of alanine (Figure 10a) and glucose (Figure 10b). Hardly any response could be seen during ischaemia and reperfusion samples in both groups, with significant differences between the pre-ischaemia and ischaemia time points. There were no significant differences between the groups. Like the epithelial separation and intestinal oxygen saturation results, the boxes of the box plot diagram appear wider in the LF group following ischaemia and reperfusion.

In both groups, the tissue conductance increased significantly during ischaemia (Figure 11). This remained elevated during reperfusion in both groups, yet with the LF group showing a slight decrease. Comparing tissue conductance between the groups, this was significantly lower in the LF group during reperfusion, with the same trend during ischaemia (*p* = 0.051).

A summary of the results is shown in Table 5, depicting the progression of the investigated variables, as well as the differences between the two ischaemia models.

## 4. Discussion

This is the first study to directly compare LF and NF ischaemia by use of sequential measurements during ischaemia and reperfusion. Based on the assumption that the type of ischaemia applied might influence the resulting IRI, the effects of LF and NF ischaemia were documented over time, and differences could be found in several of the tested variables. During ischaemia, both groups could be distinguished in serosal discolouration and wall thickness, which was also reflected histologically by the greater degree of mucosal haemorrhage in the LF group. Epithelial separation, as main measure for mucosal injury, did not differ substantially between the groups. Only after 2 hours of ischaemia did the NF group show more consistent severe epithelial separation, yet this difference did not prove to be statistically significant. Evaluating mucosal function, transcellular epithelial transport was equally affected by both ischaemia types; however, the paracellular integrity was more severely impaired following NF ischaemia. Therefore, the first hypothesis that NF ischaemia would elicit more severe mucosal damage during ischaemia could partially be accepted.

During ischaemia, increased inflammatory cell infiltration was demonstrated in the LF group compared to the NF group, but during reperfusion this did not differ between the two groups. Inflammation and apoptosis increased during reperfusion, yet histomorphologically there was no evidence of progression of intestinal damage during reperfusion. Therefore, the second hypothesis that LF ischaemia would exhibit increased inflammation and that there would be significant reperfusion injury could only partially be confirmed.

Intestinal oxygen saturation, epithelial separation, and transcellular transport results showed a relatively high variability between the horses in the LF group. Furthermore, the intestinal oxygen saturation of the different sub-segments within a single ischaemic segment was highly variable during LF ischaemia. Consequently, the hypothesis that LF ischaemia would be associated with a higher variability of the intestinal blood supply within the ischaemic segment could been confirmed.

The results of the StO_2_ indicate that there was a greater reduction in intestinal perfusion in the NF ischaemia compared to the LF ischaemia. This supports the presence of an inequality in blood flow between the two models, even though there was no difference in the blood flow measurement between the groups. Movement of the intestine during measurement by LDF affects the measured Doppler shift and causes measurement artefacts, possibly explaining why no difference could be detected between the groups. Furthermore, the higher blood flow in the low flow group during the induction of ischaemia could be associated with these movement artefacts due to ischaemia-associated hypermotility of the intestine. The increase in blood flow during reperfusion indicates that complete perfusion of the intestine occurred with the release of the ligatures. The increase in blood flow above the baseline level and the elevated Hb in the NF group during reperfusion are most likely a sign of post-ischaemic hyperaemia.

The high CV of around 50% in the LF group suggests a high variability between the different horses in StO_2_ with this type of ischaemia. The CV values of the NF horses lie below 25%, which is generally considered to be an acceptable level of variability. Contrarily, the coefficient of variation was comparably high in both groups. This may be explained by the low mean (7.1%) and standard deviation (3.7%) in the NF group, leading to a relatively high CV. Therefore, the CV may not be a clinically relevant parameter in this case, considering the low StO_2_ percentage may not have major implications for the intestinal tissue oxygenation. On the other hand, a range of 40.3% StO_2_ as seen between the different sub-segments in the LF group could be expected to affect the intestinal oxygenation and subsequent IRI. This should be kept in mind as a possible confounding factor for the interpretation of experimental trials implementing low flow ischaemia when the sequential samples are taken from different arcades at different time points.

Looking more closely at the LF ischaemia implemented in the current study, the characteristic changes of haemorrhagic ischaemia could be noted, with increasing oedema and severe purple discolouration of the intestinal surface [8]. This is most likely the result of a discrepancy in arterial and venous blood flow with venous congestion, which would not have been present in the NF group. This is also supported by the increased intestinal haemoglobin content combined with decreased blood flow, as well as the severe histologically detectable mucosal haemorrhage, most likely the result of venous congestion.

Even though blood flow measurements did not differ significantly between the groups, probably due to motion artefacts in the laser Doppler fluxmetry, the specimens of the NF group showed the typical clinical picture of NF ischaemia with a paler discolouration and less thickening of the intestinal wall [8]. Moreover, the significantly lower intestinal stO2 in the NF group confirms that there must have been a significant difference in perfusion between the groups. Nevertheless, in some of the horses subjected to NF ischaemia, an increase in intestinal wall thickness as well as mild purple discolouration could be noted during ischaemia, which would not be an expected finding if all arteries had been occluded completely. In these horses, minor mesenteric arteries that were only compressed in the mass ligature and not occluded by additional haemostatic clamps may have caused a limited influx of blood without significantly affecting intestinal tissue saturation.

The higher intestinal oxygen saturation in the horses subjected to LF compared to NF ischaemia did not translate to less severe damage in the LF group in all tested variables. Despite this group difference, the oxygen level in the LF group was nonetheless decreased compared to pre-ischaemia, which could suffice to disturb cellular mechanisms. Furthermore, the severe mucosal haemorrhage following LF ischaemia disrupting the mucosal architecture could contribute to loss of certain functions [3,8], which would not be present subsequent to NF ischaemia. Relating this to the results of the Ussing chamber experiments, the paracellular integrity was less severely affected in the LF group, indicating a higher resilience for hypoxia than the oxygen-dependent transcellular transport, which was equally affected by both ischaemia types.

A highly relevant result for the future use of these ischaemia models is the variability found following LF ischaemia. However, it should not be neglected that the NF model also exhibited noticeable differences between individual horses. Biological variation in this heterogenic population could account for this, yet this would not explain the relatively high level of variability in the LF group compared to the NF group. This could be caused by an inconsistent ischaemia grade resulting from movement artefacts in the blood flow measurements compromising the exact reduction of blood flow to 20%. On the other hand, partial vessel occlusion will be prone to a variation in blood flow, independent of the blood flow measurement accuracy at the time of ischaemia induction or the method of occlusion. Many factors such as systemic arterial blood pressure and the position of the intestines in relation to the mesenteric root will influence the blood flow through partially obstructed vessels. This was also illustrated by the high variability in intestinal oxygen saturation between the neighbouring sub-segments within a single segment during LF ischaemia, indicating differing degrees of blood flow reduction in the different jejunal arcades. This might be caused by the mass ligature eliciting varying pressures on the individual jejunal vessels. In the NF group, more homogeneity within one ischaemic segment was observed, which was to be expected considering the mass ligature was supplemented by complete haemostatic clamping of the individual vessels. This observation should be considered for experimental protocols where sequential samples are taken from neighbouring arcades subjected to LF ischaemia. Furthermore, it stresses the importance of measuring blood flow in all arcades when applying LF in a larger intestinal segment.

According to the current study, the use of a NF model appears to elicit more consistent results due to the simpler and more reproducible methodology, which is particularly suitable for experimental studies with a low number of subjects. Notwithstanding, it remains a point of discussion whether LF ischaemia corresponds more closely to the pathophysiologic process of strangulation of the equine small intestine. A possible alternative for the complete or partial occlusion of both venous and arterial vessels is the complete occlusion of the mesenteric veins, as has been used in previous studies [3,8]. This model will be accompanied by severe intramural haemorrhage as seen in clinical patients, yet it will probably also show a certain degree of variability depending on the systemic blood pressure and intramural pressure. Furthermore, it is questionable if a complete lack of arterial occlusion is representative for the clinical situation. At this point, this cannot be clarified conclusively, because this is not supported by blood flow measurements in clinical patients.

Evaluating the immunohistochemistry results, more inflammatory cell infiltration could be noted in the LF group during ischaemia, most likely the result of the partial vessel occlusion during LF that still allowed circulating inflammatory cells to reach the intestinal segment. During reperfusion, there was more inflammatory cells infiltration in the submucosa in the NF group compared to the LF group. This could be the result of more severe intestinal damage in the NF group, resulting in greater infiltration of the intestine with circulating inflammatory cells. In both groups, there was only a very slight increase in apoptotic cells during ischaemia. After 1 h and 2 h of reperfusion, the number of cleaved caspase-3 positive cells increased significantly in both groups. This response appeared more pronounced after NF ischaemia, yet without statistically significant differences between the groups.

As mentioned previously, the progression of mucosal damage during reperfusion in LF and NF models has been a point of conflicting results in previous experimental studies. Generally, LF studies more commonly reported reperfusion-related injury [5,9,12]. In the current study this could be found neither for mucosal histomorphology nor for epithelial function. Looking closer at the epithelial separation scores (Figure 7a), it does seem that the horses in the LF group that had less severe damage during ischaemia seem to catch up with the other horses during reperfusion, with all horses showing similar scores and less variability at the end of the experiment. This would support the theory that if only mild injury is incurred during ischaemia, further damage during reperfusion is more likely [9]. However, we cannot support this with statistical analysis due to the small number of horses. Other parameters for IRI are the amount of inflammatory and apoptotic cells. We found significant inflammatory cell infiltration and increase in apoptotic cells in both groups during reperfusion.

This raises the question of whether apoptosis and inflammatory cell infiltration is a response to the damage elicited by the oxygen deficiency during ischaemia or a consequence of oxidative insults due to the formation of reactive oxygen metabolites during reperfusion. Comparing the time frames of previously performed studies by the same group using the same markers for apoptosis, the induction of apoptosis does not appear to be dependent on the time counting from the start of ischaemia, but related to the start of reperfusion [5,7,15]. This suggests that this apoptosis is induced as a result of reperfusion itself, which could be interpreted as reperfusion injury. The influx of inflammatory cells as indicators for reperfusion injury should be interpreted with more caution, considering this can only be seen after the blood flow is reinstated.

Regarding the transmucosal transport, hardly any response in short circuit current following alanine and glucose addition could be seen during ischaemia and reperfusion samples in both groups. This indicates that the sodium-dependent transcellular transport of alanine and glucose was affected by ischaemia, and this had not recovered during reperfusion. There were no significant differences between the groups, suggesting that both types of ischaemia equally impeded transcellular transport. Comparing tissue conductance between the groups, this was significantly lower in the LF group during reperfusion, with the same trend during ischaemia. This indicates that the paracellular integrity of the epithelium was not as severely affected by LF ischaemia as it was by NF ischaemia.

Quantification of blood flow by Laser Doppler fluxmetry can be used to determine the local microcirculation in the intestinal wall [5,6,15,28], which is an advantage compared to measuring the macrocirculation based on Doppler sonography [12,17,19] or via blood pressure measurement [9] in the mesenteric blood vessels. Nevertheless, the intestinal hypermotility seen following the induction of ischaemia led to severe motion artefacts in Laser Doppler fluxmetry measurements. Consequently, the implementation of LF ischaemia based on this measurement method did not allow the exact reduction of blood flow by the desired amount, representing a limitation of this model. Intestinal oxygen saturation measurement by microlightguide spectrophotometry was not as susceptible to movement artefacts and appeared to be a more reliable method to validate the type of ischaemia. As it does not respond directly to the reduction in blood flow, it is less suitable as an immediate control value for creating a defined low flow ischaemia, yet it can be used for sequential measurements.

A limitation of the implemented ischaemia-inducing technique is the use of a mass ligature. As mentioned previously, this could have caused differing pressures on the individual vessels in the LF group, in which this was not supplemented by haemostatic clamps. Furthermore, the amount of fat within the mesentery can influence the pressure that is exerted on the vessels. Nevertheless, the reason for using this type of vessel occlusion was to simulate the clinical situation of intestinal strangulation more accurately, where the mesentery is compressed and usually not the individual vessels. Furthermore, this mass ligature ensures compression of all vessels, even the smaller ones that could be overlooked in a fatty mesentery. For NF ischaemia, the addition of haemostatic clamps seems a secure option, yet this does not allow partial compression. An alternative for LF ischaemia could be the addition of partially occluding vascular clamps or ligatures around the individual arcades.

Other limitations of the study include the small number of horses per group, which may have contributed to the fact that there were less significant differences between the groups due to the unforeseen high variability in the low flow group. The relatively short time frame of the experimental trial is a limitation for the comparison between the experimentally-induced ischaemic injury and that of clinical cases with intestinal strangulations. Furthermore, pharmacologic preconditioning by the anaesthetics may have attenuated ischaemia- or reperfusion-induced injury. However, the latter was present in both groups and should not have affected the group comparison.

## 5. Conclusions

The results of this study illustrate the effects of LF and NF ischaemia on the equine small intestine. Repeated sampling over time provided extensive information on the progression of the different parameters for intestinal damage during ischaemia and reperfusion, which may be used to delineate the timing of future studies. Furthermore, the comparison between the two models offers backing for the choice of the appropriate experimental model. NF ischaemia demonstrated less variability in StO_2_ and elicited tissue damage, which suggests that this model may be more suitable for an experimental setting with fewer subjects where consistent ischaemia is essential. On the other hand, LF ischaemia was associated with severe intramural haemorrhage and therefore may be more appropriate to simulate certain causes of clinical intestinal ischaemia. The evaluation of blood flow in clinical cases of small intestinal strangulations is necessary to clarify which model is most representative for the clinical situation. Possibly, a blended form of ischaemia combining complete venous occlusion for a shorter time frame followed by additional complete arterial occlusion could elicit both clinically relevant intramural haemorrhage as well as a more standardized ischaemia.

## Figures and Tables

**Figure 1 animals-12-02158-f001:**
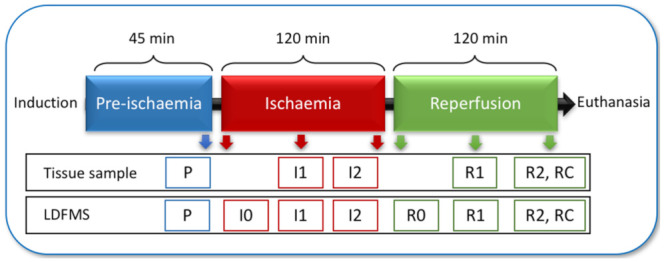
Timeline of the experiment and associated sampling. P = pre-ischaemia; I0 = immediately following induction of ischaemia; I1 = after 60 min ischaemia; I2 = after 120 min of ischaemia; R0 = immediately following reperfusion; R1 = after 60 min of reperfusion; R2 = after 120 min of reperfusion; RC = remote control segment after reperfusion.

**Figure 2 animals-12-02158-f002:**
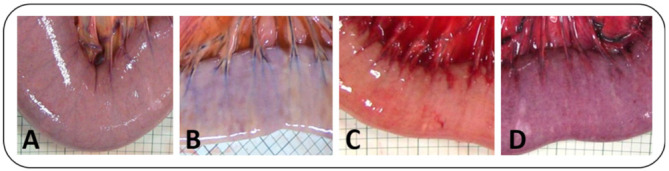
Exemplary images of different grades of intestinal discolouration following ischaemia and reperfusion.

**Figure 3 animals-12-02158-f003:**
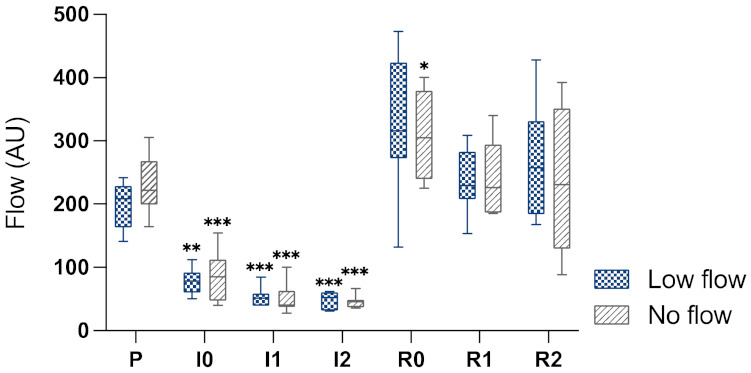
Box-plot diagram of the intestinal microperfusion measured by LDS in arbitrary units (AU) during the different time points. P, pre-ischaemia; I0, immediately after induction of ischaemia; I1, 60 min ischaemia; I2, 120 min ischaemia; R0, immediately following reperfusion; R1, 60 min reperfusion; R2, 120 min reperfusion. The horizontal bar displays the median, the interquartile range is represented by the box, and the minimum and maximum by the whisker plots. Significant differences of the time points compared to pre-ischaemia are marked with an asterisk (* *p* < 0.05; ** *p* < 0.01; *p* *** < 0.001). There were no significant differences between the groups.

**Figure 4 animals-12-02158-f004:**
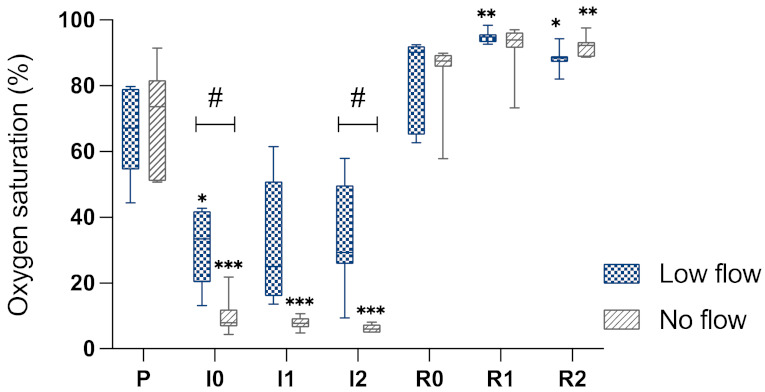
Box-plot diagram of the intestinal StO_2_ by MS in% during the different time points. P, pre-ischaemia; I0, immediately after induction of ischaemia; I1, 60 min ischaemia; I2, 120 min ischaemia; R0, immediately following reperfusion; R1, 60 min reperfusion; R2, 120 min reperfusion. Significant differences of the time points compared to pre-ischaemia are marked with an asterisk (* *p* < 0.05; ** *p* < 0.01; *p* *** < 0.001), significant differences between the groups are indicated with a horizontal bar and accompanying hash (# *p* < 0.05).

**Figure 5 animals-12-02158-f005:**
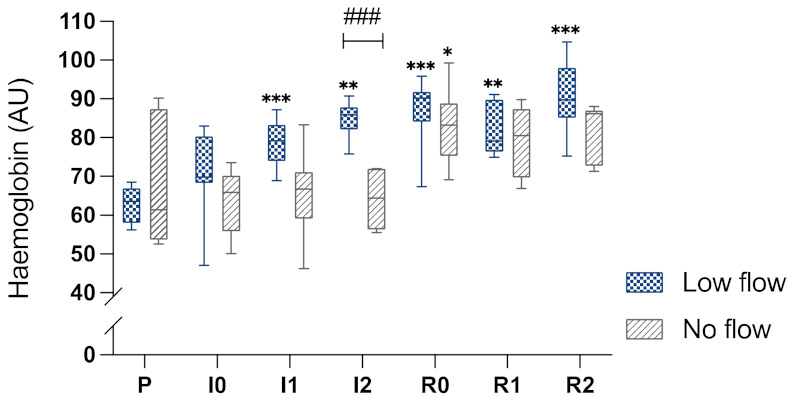
Box-plot diagram of the intestinal haemoglobin measured by MS in arbitrary units (AU) during the different time points. P, pre-ischaemia; I0, immediately after induction of ischaemia; I1, 60 min ischaemia; I2, 120 min ischaemia; R0, immediately following reperfusion; R1, 60 min reperfusion; R2, 120 min reperfusion. Significant differences of the time points compared to pre-ischaemia are marked with an asterisk (* *p* < 0.05; ** *p* < 0.01; *** *p* < 0.001), significant differences between the groups are indicated with a horizontal bar and accompanying hash (### *p* < 0.001).

**Figure 6 animals-12-02158-f006:**
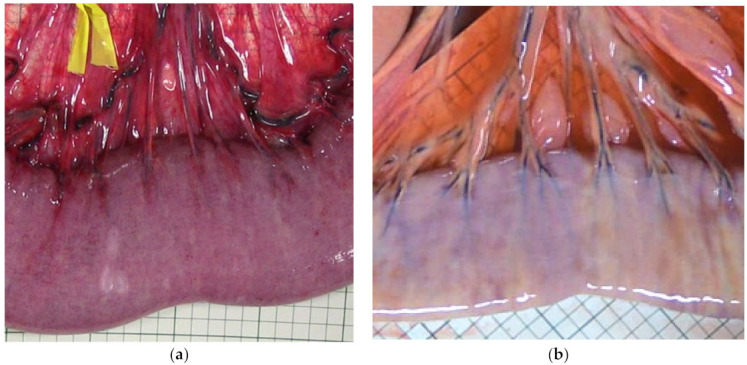
Clinical appearance of the jejunum after being subjected to two hours of different segmental ischaemia: low flow ischaemia (**a**) displaying intense purple discolouration, and no flow ischaemia (**b**) with mild purple discolouration and vessel-associated pale areas.

**Figure 7 animals-12-02158-f007:**
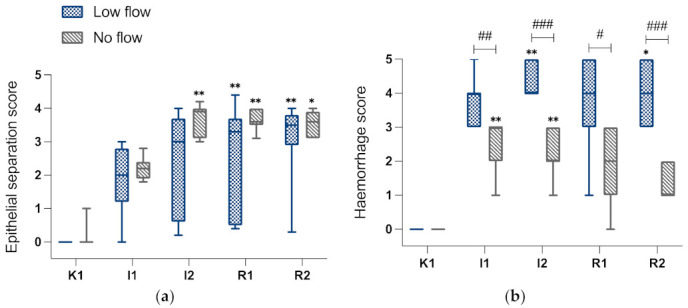
Box-plot diagrams of EPS (**a**) and mucosal HS (**b**) evaluated according to modified Chiu-score of the LF and NF group for the different intestinal samples. P, pre-ischaemia; I1, 60 min ischaemia; I2, 120 min ischaemia; R1, 60 min reperfusion; R2, 120 min reperfusion. Significant differences of the time points compared to pre-ischaemia are marked with an asterisk (* *p* < 0.05; ** *p* < 0.01), significant differences between the groups are indicated with a horizontal bar and accompanying hash (# *p* < 0.05; ## *p* < 0.01; ### *p* < 0.001).

**Figure 8 animals-12-02158-f008:**
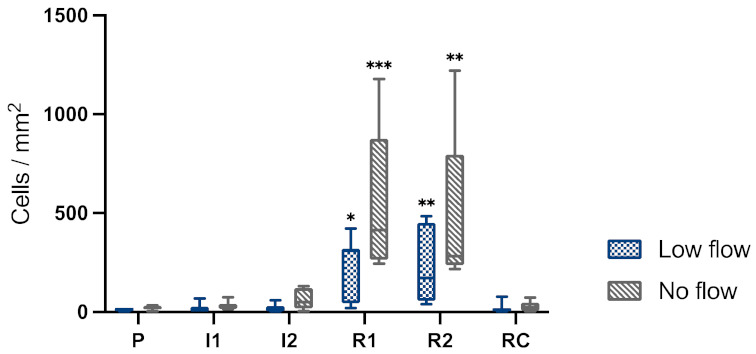
Box-plot diagram of the cleaved caspase-3-positive cells/mm^2^ in the mucosa of jejunum subjected to either LF or NF ischaemia at different time points. P, pre-ischaemia; I1, 60 min ischaemia; I2, 120 min ischaemia; R1, 60 min reperfusion; R2, 120 min reperfusion; RC, remote control sample at reperfusion. Significant differences are marked with an asterisk (* *p* < 0.05; ** *p* < 0.01; *** *p* < 0.001).

**Figure 9 animals-12-02158-f009:**
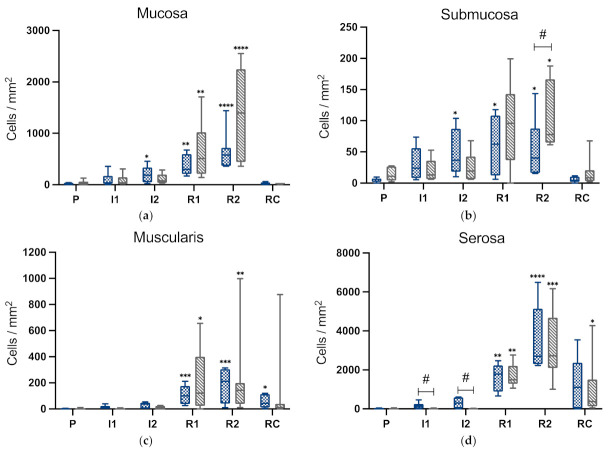
Box-plot diagram of positive cell counts in cells/mm^2^ after immunohistochemical staining for cytosolic calprotectin in the mucosa (**a**), submucosa (**b**), muscularis (**c**) and serosa (**d**) of jejunum subjected to either LF or NF ischaemia at different time points. P, pre-ischaemia; I1, 60 min ischaemia; I2, 120 min ischaemia; R1, 60 min reperfusion; R2, 120 min reperfusion; RC, remote control sample at reperfusion. Significant differences of the time points compared to pre-ischaemia are marked with an asterisk (* *p* < 0.05; ** *p* < 0.01; *** *p* < 0.001; **** *p* < 0.0001), significant differences between the groups are indicated with a horizontal bar and accompanying hash (# *p* < 0.05).

**Figure 10 animals-12-02158-f010:**
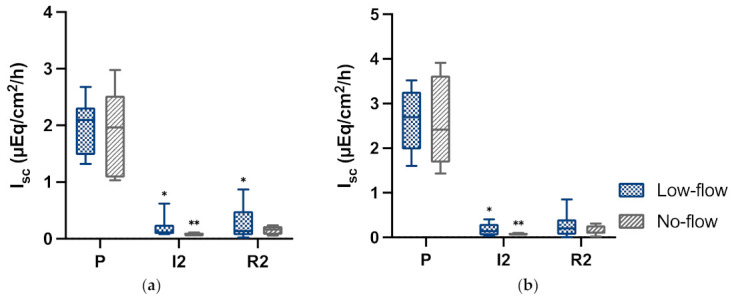
Box-plot diagrams of the short circuit currents (I_sc_) following the addition of (**a**) Alanine; or (**b**) Glucose during pre-ischaemia (P), following 120 min of ischaemia (I2) and after 120 min of reperfusion (R2) in horses subjected to low and NF ischaemia. The horizontal bar displays the median, the interquartile range is represented by the box, and the minimum and maximum by the whisker plots. Significant differences of the time points compared to pre-ischaemia are marked with an asterisk (* *p* < 0.05; ** *p* < 0.01).

**Figure 11 animals-12-02158-f011:**
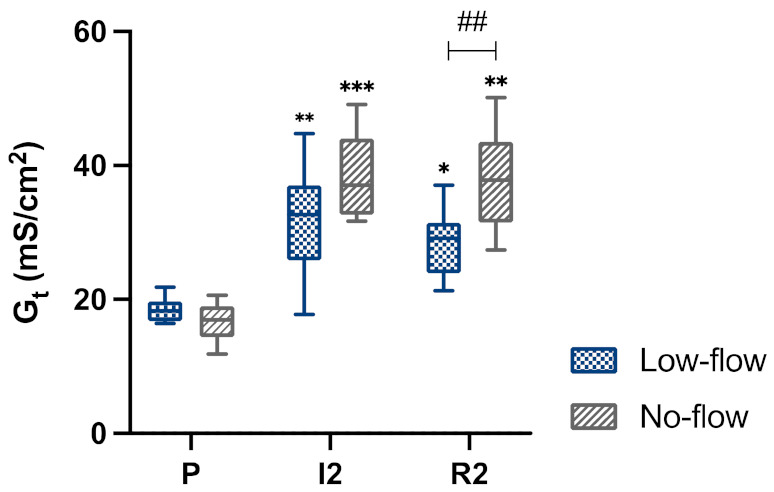
Box-plot diagram of the tissue conductance (G_t_) during pre-ischaemia (P), following 120 min of ischaemia (I2) and after 120 min of reperfusion (R2) in jejunum subjected to either LF or NF ischaemia. Significant differences of the time points compared to pre-ischaemia are marked with an asterisk (* *p* < 0.05; ** *p* < 0.01; *** *p* < 0.001), significant differences between the groups are indicated with a horizontal bar and accompanying hash (## *p* < 0.01).

**Table 1 animals-12-02158-t001:** Semiquantitative scores for the clinical evaluation of the intestine.

Grade	Wall Thickness	Grade	Intestinal Discolouration
0	Physiologic (3 mm)	A	Physiological—light pink
1	Mild thickening (3–5 mm)	B	Pale to light purple with pale patches
2	Moderate thickening (5–10 mm)	C	Dark pink to red
3	Severe thickening (10–15 mm)	D	Dark Purple

**Table 2 animals-12-02158-t002:** Modified Chiu score * for histomorphology of intestinal mucosa.

Grade	Epithelial Separation	Haemorrhage
0	Normal mucosal villi	None
1	Slight separation of epithelial cells from the lamina propria at the tip of the villus (Gruenhagen’s space)	Few extravascular individual red blood cells
2	Extension of subepithelial space ± loss of epithelial cells from the tip of the villus	Mild local haemorrhage
3	Extension of the subepithelial space with epithelial lifting down the sides of the villi exposing a third to a half of the lamina propria	Mild diffuse haemorrhage ± moderate local haemorrhage (no clumping of the red blood cells)
4	Complete separation of epithelium from lamina propria to the villus base (denuded villi)	Moderate diffuse haemorrhage ± severe local haemorrhage, including local clumping of red blood cells
5	Loss of villus architecture and early necrosis of the crypt cells	Massive haemorrhage

* Adapted from Chiu 1970 [25]; Verhaar et al. 2021 [6].

**Table 3 animals-12-02158-t003:** StO_2_ range and CV of the ischaemia in the intestinal sub-segments within one ischaemic segment of a single horse averaged per group.

Ischaemia Model	Range of StO_2_ between Sub-Segments within a Single Horse (Mean of All Horses)	CV of StO_2_ between Sub-Segments within a Single Horse (Mean of All Horses)
LF	40.3%	48.2%
NF	10.0%	51.9%

**Table 4 animals-12-02158-t004:** Frequency distribution of intestinal wall thickness and discolouration of jejunum subjected to LF and NF ischaemia.

	Group	Wall Thickness ^2^					Serosal Discolouration ^3^			
Score		0	1	2	3		A	B	C	D
Pre-ischaemia	LF	7					7			
	NF	7					7			
1 h ischaemia ^a,b^	LF			5	2			2	1	4
	NF		5	1	1			7		
2 h ischaemia ^a,b^	LF			1	6			2	1	4
	NF		3	3	1			7		
1 h reperfusion	LF		1	3	3				7	
	NF	2	2	2	1				7	
2 h reperfusion ^a^	LF		2	4	1				7	
	NF	2	5						7	

Significant differences between the LF and NF groups at the individual time points are marked with an ‘a’ in case of difference in wall thickness and ‘b’ for serosal discolouration (*p* < 0.05). ^2^ Intestinal wall thickness grade: 0—physiologic (3 mm); 1—mild thickening (3–5 mm); 2—moderate thickening (5–10 mm); 3 severe thickening (5–15 mm). ^3^ Serosal discolouration grade: A—light/pale pink; B—pale to light purple with pale patches; C—dark pink—red; D—dark purple.

**Table 5 animals-12-02158-t005:** Summary of the results comparing the progression of LF and NF ischaemia. LDFMS Laser Doppler fluxmetry and microlightguide spectrophotometry; I Ischaemia; R Reperfusion; = no (further) changes; ↓ mild/↓↓ moderate/↓↓↓ severe decrease; ↑ mild/↑↑ moderate/↑↑↑ severe increase; nss not statistically significant.

			Low Flow (LF)	No Flow (NF)	Group Comparison
**LDFMS**	*Intestinal blood flow*	**I**	↓↓↓	↓↓↓	No difference between groups
		**R**	↑↑↑	↑↑↑	No difference between groups
	*Intestinal oxygen saturation*	**I**	↓↓	↓↓↓	LF > NF LF: higher variation
		**R**	↑↑	↑↑	No difference between groups
	*Intestinal haemoglobin*	**I**	↑↑↑	=	LF > NF
		**R**	=	↑	No difference between groups
**Clinical observations**	*Serosal discoloration*	**I**	Dark purple discolouration	Pale purple with white patches	Colour difference
		**R**	Dark pink–red	Dark pink–red	No difference between groups
	*Wall thickness*	**I**	↑↑↑	↑	LF > NF
		**R**	=	=	No difference between groups
**Mucosal histomorphology**	*Epithelial separation*	**I**	↑↑	↑↑↑	LF < NF (nss)
		**R**	=	=	No difference between groups
	*Haemorrhage*	**I**	↑↑↑	↑	LF > NF
		**R**	=	=	LF > NF
**Immunohisto-** **chemistry**	*Inflammation*	**I**	↑	↑	LF > NF (Serosa)
		**R**	↑↑↑	↑↑↑	LF < NF (Submucosa)
	*Apoptosis*	**I**	↑	↑	No difference between groups
		**R**	↑↑↑	↑↑↑	LF < NF (nss)
**Electrophysiology**	*Short circuit* *currents*	**I**	↓↓↓	↓↓↓	No difference between groups
		**R**	=	=	No difference between groups
	*Tissue conductance*	**I**	↑↑	↓	No difference between groups
		**R**	=	=	LF < NF

## Data Availability

The data presented in this study are openly available in Mendeley Data under the following link http://dx.doi.org/10.17632/3hkh8v9mvv.1.

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
