# Peer review of "Low Flow versus No Flow: Ischaemia Reperfusion Injury Following Different Experimental Models in the Equine Small Intestine"

_animals, 2022, doi:10.3390/ani12162158_

Round 1
Reviewer 1 Report
General comment:
Overall, a very well written and interesting manuscript. The experiments performed were extremely thorough and nicely put together. The comparison of no flow to low flow ischemia contributes significantly to the interpretation of previous studies as well as contributes to more effective planning for future experiments that contribute to understanding this form of intestinal injury. Interpretation of results are present within the results section and should be removed and limited to the discussion section. A reformatting of the graphs through (As commented on specifically below) should be considered. Finally, a figure is needed to demonstrate the appearance of tissues following LF and NF ischemia.
Specific Comments
Abstract:
Line 35-36: Methodologically, full thickness biopsies are not commonly used for Ussing chamber analysis, it is usually just the mucosa that is separated from the underlying layers and then mounted on the chambers. This provides information regarding transepithelial resistance and flux (if measured), please clarify.
Introduction:
Lines 48-64: There appears to be a lot of redundancy in what is written. I believe that this can be made more concise and clear.
Line 79: The successful culture of equine organoids has been published and should be accurately referenced here. Equine monolayer culture has also been published.
Materials and Methods
Line 135-136: please clarify, until ‘arterial’ blood flow was reduced
Section 2.3: The serosal color grade does not seem to take into account the appearance of tissue that has complete vascular occlusion and this usually results in a completely pale appearance. It definitely should not be equivalent to a normal piece of intestine in appearance. Please comment on this.
Section 2.5: The methods of separating the mucosa from remaining tissue is accurately described here, but in the abstract it is misleading. Please accurately represent what tissue was evaluating in the Ussing chamber in the abstract.
Line 205: Is there a reason that alanine or glucose were used and that this was not kept consistent?
Lines 212-215: Wonderful to see a power analysis performed. However in the calculation does it indicate the 14 animals per NF or LF? Or do the authors mean 7 in each group? I would suspect that the power analysis for detecting statistical significance in the measurement of apoptotic cells, for example, and Ussing chamber work would have likely been different. Did this number represent that minimum number for all proposed benchtop experiments? Please clarify.
Results:
Lines 254-261: These lines are discussion/conclusions based on the results and should be appropriately moved to that section. The results should clearly state what was found and then be interpreted in the discussion.
Figure 3: The representation of results is a little confusing since both the comparison to pre-ischemia and between LF and NF are being done in the same graph. Would it be better to graph the change between pre-ischemia and each subsequent time point and then compare LF to NF. The objective the manuscript is to actually compare LF to NF. The text could then state that both methods effectively reduced the flow (or increased the flow during reperfusion).
Figure 4: As commented on in Figure 4, perhaps use different symbols to indicate when comparisons are performed between baseline and those between LF and NF?
Lines 276-282: This paragraph is an interpretation of the results which should be in the discussion not in the results.
Line 298: typo, sentence starting with Therefore, the CV may not ‘be’ a clinically relevant…
Lines 296-303: interpretation of results that should be in discussion not results section.
Lines 315-318: interpretation of results that should be in discussion not results section.
Figure 5: Typo is spelling of Hemoglobin on y axis
Table 4: The change in serosal colour between the groups is misleading as the appearance of the truly NF vs LF is completely different. I would consider modifying the colour discoloration scoring. The appearance of NF (if truly NF) should be very pale with a perhaps mild purple discoloration, which is very different from the dark purple and congested appearance of LF. I also find it very strange that within 2 hours of NF that there was a change of wall thickness in the NF group. This seems to indicate that perhaps the NF group was not completely occluded and this should be commented on (as this can happen and is not totally unexpected in this model despite clamping).
Figure 6: A lot of comparisons are being made within the same graphs. I would recommend demonstrating the most salient points (the most important comparisons) on the graph and then commenting on the rest in the text.
Comment: A statement in the methods should be added to indicate what calprotectin staining is identifying. What cells are expressing calprotectin and why.
Line 397: How were neutrophils specifically being identified?
Lines 395-401: interpretation of results within this paragraph which should be moved to the discussion.
Lines 415-421: interpretation of results
Discussion:
Overall, very nicely written. A comment could be added regarding the interpretation results based on a model of only 2 hours of ischemia as in clinical cases the duration of ischemia is likely much longer in most cases. In addition the authors do not comment on the possible role of physical disruption of the tissue caused by red blood cells when tissue is engorged following venous occlusion. This type of injury is not present in no flow tissue.
Reviewer 2 Report
the authors described the effects of two models of experimentally- induced ischemia in alive horses. The topic is very interesting and has a high welfare importance in my opinion. The results may in fact help in adopting the best experimental model in IRI studies, thus reducing the number of animal to be scarified.
the study is well described in all its part. the results are a little to long but clearly described. table 5 gives an excellent picture of the results .
I only have some points that need to be highlighted better in the paper.
How did you evaluate intestinal wall thickness? please add details about this
table 4: if I am not wrong these are categorical data. Instead of median and range, I think that it would have been more appropriate to describe frequency distribution.
line 202-203: I don't understand the meaning of this sentence. please reword.
as the authors said, sistyemic blood pressure may have influenced come of the variable in the LF group. do you evaluate systemic blood pressure and blood gas analysis accordingly? it would be nice if the authors could add some information about it. Were the horses mechanically ventilated?
As the authors stated LF group showed greater variability but I agree that this condition is closer to reality .
Reviewer 3 Report
Dear Authors, please correct the table 1 in connection with table 4! Check the statistical analysis too!

Round 2
Reviewer 2 Report
I am pleased to find that the authors replied to all the comments and suggestions I gave them.
I feel that the paper is more complete now than before and suitable. for pubblication.
Thank you
Author Response
Thank you for your constructive comments and contribution to our paper!